# Antibacterial Effects of a Carbon Nitride (CN) Layer Formed on Non-Woven Polypropylene Fabrics Using the Modified DC-Pulsed Sputtering Method

**DOI:** 10.3390/polym15122641

**Published:** 2023-06-10

**Authors:** Young-Soo Sohn, Sang Kooun Jung, Sung-Youp Lee, Hong Tak Kim

**Affiliations:** 1Department of Biomedical Engineering, Daegu Catholic University, Gyeongsan 38439, Republic of Korea; sohnys@cu.ac.kr; 2Econet Korea Ltd., Gumi 39373, Republic of Korea; econetkorea@naver.com; 3Department of Physics, Kyungpook National University, Daegu 41566, Republic of Korea; physylee@knu.ac.kr

**Keywords:** antibacterial effect, carbon nitride, staphylococcus aureus, klebsiella pneumonia, non-woven fabric, sputtering

## Abstract

In the present study, the surface of non-woven polypropylene (NW-PP) fabric was modified to form CN layers using a modified DC-pulsed (frequency: 60 kHz, pulse shape: square) sputtering with a roll-to-roll system. After plasma modification, structural damage in the NW-PP fabric was not observed, and the C–C/C–H bonds on the surface of the NW-PP fabric converted into C–C/C–H, C–N(CN), and C=O bonds. The CN-formed NW-PP fabrics showed strong hydrophobicity for H_2_O (polar liquid) and full-wetting characteristics for CH_2_I_2_ (non-polar liquid). In addition, the CN-formed NW-PP exhibited an enhanced antibacterial characteristic compared to NW-PP fabric. The reduction rate of the CN-formed NW-PP fabric was 89.0% and 91.6% for *Staphylococcus aureus* (ATCC 6538, Gram-positive) and *Klebsiella pneumoniae* (ATCC4352, Gram-negative), respectively. It was confirmed that the CN layer showed antibacterial characteristics against both Gram-positive and Gram-negative bacteria. The reason for the antibacterial effect of CN-formed NW-PP fabrics can be explained as the strong hydrophobicity due to the CH_3_ bond of the fabric, enhanced wetting property due to CN bonds, and antibacterial activity due to C=O bonds. Our study presents a one-step, damage-free, mass-productive, and eco-friendly method that can be applied to most weak substrates, allowing the mass production of antibacterial fabrics.

## 1. Introduction

Non-woven (NW) fabrics are fabric-like materials made of short and long polymer fibers, which are held together with chemical, mechanical, and thermal treatments. Polypropylene (PP) is currently the most popular material to manufacture NW fabrics due to its low cost, light weight, good chemical stability, excellent moisture resistance, and insulation properties. In addition, PP is an eco-friendly material that is naturally decomposed outdoors. NW fabrics, including PP, have been widely used in various applications, such as medical supplies, food packing materials, sanitary products, fashion textiles, civil engineering products, and building materials [1,2,3]. In principle, the materials used for biomedical applications should be biocompatible and have antibacterial properties. Pathogens such as bacteria and viruses cause fatal injuries to the human body, and if they spread widely, they cause great social and economic losses. Until now, removing bacteria and viruses has been considered a problem to be solved in the category of medicine and pharmacy. Recently, it has been reported in various research institutes around the world that virus removal can be performed even by using the antibacterial properties of metals known for a long time. Typically, antibacterial metals, such as silver, platinum, and copper, are commonly applied to various materials in the form of nanomaterials and thin films [4,5,6]. The basic mechanisms of antibacterial activity for antibacterial metals and metal oxides are the enzyme interference of metal ions, the production of reactive oxygen species (ROS), the destruction of cell membranes, the direct genotoxic activity for some metals, and the prevention of absorption of important microelements by microbes [7]. However, these metals can be easily oxidated, and their antibacterial effect can be rapidly reduced. Metal oxides, including ZnO nanoparticles, can promote the generation of ROS, and this can lead to antibacterial effects [7,8]. However, studies on the harmfulness of nanoparticles to the human body are needed, and the mass production of nanoparticles is also difficult. Compared to these materials, carbon and carbon-related materials are relatively safe for the human body and have the advantage of being inexpensive. Recently, antibacterial effects of carbon-related materials, such as diamond-like carbon, diamond, and graphitic carbon nitride (g-C_3_N_4_), have been investigated, and several studies confirmed their antibacterial effect [9,10,11,12,13]. For example, activated carbon (AC) has a larger surface area due to its porous structure, and this property can efficiently promote antibacterial effects. In addition, antibacterial effects can also be promoted if we manipulate the surface of these materials. More specifically, super-hydrophilic and -hydrophobic surfaces have been reported to prevent bacterial attachment to surfaces, resulting in low bacterial adhesion [9,14,15]. However, these properties are difficult to achieve on weak materials such as fabrics, NW fabrics, paper, and textiles. Previous studies have shown that wettability can be controlled through carbon surface treatment on paper, which is a soft and porous substrate, and it has been confirmed that there is no damage to the paper [16]. This study investigated the antibacterial effect induced by the formation of a carbon-nitride (CN) layer on the NW-PP fabric. The CN layer was formed on the NW-PP fabrics using a modified DC-pulsed sputtering, which has been shown to minimize substrate damage [16,17]. The main strategy for acquiring the antibacterial effect was to use the porous structure of NW-PP fabrics and modify their surfaces to convert a C–C bond to a C–N bond. It is expected that this design can increase the surface area and promote the reaction with pathogens.

## 2. Materials and Methods

The NW-PP fabrics were treated by DC-pulsed magnetron sputtering. The sputtering device is equipped with a roll-to-roll device, suitable for the flexible substrate process. The configuration of the magnet was alternately repeated N-polarity lines and S-polarity lines. The magnet arrays of the sputtering gun were configured to acquire a magnetic field parallel to the substrate using magnetic simulation software (FEMM) [18]. The size of the carbon target was 0.416 m × 0.298 m, and the distance between the target and the NW-PP fabrics was 6 cm. The chamber was evacuated up to a base pressure of 6.7 × 10^–4^ Pa before the modification process of the NW-PP fabrics, and the working pressure was maintained at 0.23 Pa (Ar: 180 sccm, N_2_: 20 sccm) during the deposition process. The plasma was generated by a DC-pulsed power (SJ power, Bucheon, Republic of Korea, SPF-2). The applied DC pulse was a negative square shape with a frequency of 60 kHz and a duty ratio of 50%. During the plasma treatment, a power of 450 W (voltage: 450 V, current: 1 A, current density: 124 mA/m^2^) was applied to the sputtering source. The CN layer formations were performed at room temperature, and the treatment time was 20 s (roll speed: 100 mm/min). The schematic diagram of the modified DC-pulsed sputtering system is shown in Figure 1. One disadvantage of sputtering deposition is a line-of-sight process, and this means that conformal deposition is difficult on complex-shaped substrates, including porous structural materials. In the modified sputtering, it is possible to widen the incident angle of particles into the substrate because of the modified sputtering source and the roll-to-roll system. Although highly uniform deposition through porous materials is not easy using this system, conformal processing is possible to a certain extent [16].

The emission spectrum of the Ar/N_2_-C plasma was measured by an optical emission spectrometer (Avantes, Gelderland, Netherlands, AvaSpec-256) with a blazed grating (groove density: 600 line/mm, blaze: 250 nm) in the range of 200–900 nm. The surface morphology of the CN-formed NW-PP fabrics was investigated using a scanning electron microscope (SEM, Hitachi, S-4800) operating at the acceleration voltage of 5 kV. The elemental distribution of the fabrics was analyzed at the acceleration voltage of 15 kV using an energy-dispersive X-ray spectroscope (EDX) with an X-ray detector (Horiba, Kyoto, Japan, X-Max^N^). The chemical bonds on the surface of the fabrics were analyzed by an X-ray photoelectron spectrometer (XPS, ThermoFisher, Waltham, MA, USA, Nexsa) using Al-K_α_ radiation (1486.6 eV). The contact angle was measured by a contact angle goniometer with a digital microscope camera (Amscope, Irvine, CA, USA, MU1000,). The test solutions were distilled water (DI, H_2_O) and diiodomethane (CH_2_I_2_, 99%, Sigma-Aldrich, St, Louis, MO, USA), which were representative polar and dispersive solutions, respectively. The contact angle was evaluated using image analysis software (ImageJ) [19], and the calculations of surface energy were performed by the Owen–Wendt method [20]. The antibacterial tests of the CN-formed NW-PP fabrics were performed by appointing an authorized agency (Korea Far Infrared Association Co., Seoul, Republic of Korea). The KS K 0693 test was employed as the antibacterial test method [21]. A brief summary of the KS K 0693 test was as follows: first, reference and fabric samples were sterilized at a high temperature of 120 °C and a high pressure of 103 kPa for 20 min before the test; second, the bacteria culture was injected into the sterilized samples; then, the samples were incubated at a temperature of 37 °C for 18 h. *Staphylococcus aureus* (ATCC 6538, *S. aureus*) and *Klebsiella pneumoniae* (ATCC 4352, *K. pneumoniae*) were used as the representative strains of Gram-positive and Gram-negative bacteria, respectively. The reduction rate (*R*) was obtained through a comparison of bacterial growth in reference and NW-PP fabrics. The reference sample for the antibacterial test was a cotton fiber, which had a mass per unit area of 115 ± 5 g/m^2^ and a whiteness index of 70 ± 5 measured by a reflectometer under a CIE standard light source of D_65_ condition (KS K ISO 105-F02:2001).

## 3. Results and Discussion

Optical emission spectroscopic analysis is a useful technique for analyzing active species in processing plasma, and Figure 2 demonstrates the optical emission spectrum of Ar/N_2_–C plasma in the DC-pulsed sputtering system. For Ar emissions, we observed neutral argon lines (Ar I) in the range of 580–850 nm and single ionized argon lines (Ar II) in the range of 400–520 nm [16,22,23]. Some Ar I lines overlapped with Ar II lines at a wavelength of 415.9, 420.1, and 427.2 nm. For N_2_ emissions, we noted that all the emission lines of N_2_ were attributed to molecular transitions and overlapped with the Ar emission lines mostly in the 400–790 nm range. The first positive lines of N_2_, between 590 and 790 nm, were attributed to the transition of B^3^Π_g_ − A^3^Σ_u_. The second positive lines of N_2_ in the range of 340 and 400 nm were ascribed as the transition of C^3^Π_u_ − B^3^Π_g_, and the first negative line of N_2_ at a wavelength of 290 nm was attributed to B^2^Π_u_^+^ − X^2^Σ_g_^+^, as shown in Figure 2 (inset) [24,25]. For carbon emissions, the atomic emission lines at 427 nm and 465 nm were attributed to a singly ionized carbon line (C II) and a doubly ionized carbon line (C III), respectively. Finally, the molecular emission band in the range of 469–474 nm was ascribed to the transition of C_2_ (d^3^Π_g_ − a^3^Π_μ_) [16]. From this result, it could be seen that the DC-pulsed magnetron sputtering equipment produced enough carbon and nitrogen active species to form the CN-contained layers on the surface of NW-PP fabrics.

Figure 3 shows SEM images of the surface morphology for ordinary NW-PP and CN-formed NW-PP fabrics and the EDX image of elemental distribution for C, N, and O for CN-formed NW-PP fabric. It is shown that there were no differences in the surface morphology between the two samples, and the fabric structure remained undamaged after plasma treatment. The weight ratios of C, N, and O were 97.23, 1.51, and 1.27%, respectively. As shown in Figure 3c, the CN layer uniformly covered the surface of the NW-PP fabric. The latter could be explained by the fact that a short plasma exposure time can cause less damage, and the pulsed power could effectively reduce the charging damage of the substrates [16,17,26]. The thickness of the CN layer formed on the fabrics was measured indirectly due to the short processing time of 20 s and the low sputtering yield of the graphite target. According to the literature, the sputtering yield of carbon was approximately 0.2 for argon ions striking the carbon target with a kinetic energy of 600 eV [16,27,28]. This value is very low compared to other target materials. The growth rate was found to be approximately 0.3 Å/s by converting the thickness after 2 min of deposition on the glass substrate. In addition, no significant differences were observed by the FT-IR and Raman measurements (not shown here). The difference between ordinary and CN-formed NW-PP fabrics was only observed in XPS measurements, which was a surface-sensitive analysis with a typical penetration depth of a few nanometers. From these results, it could be estimated that the CN layer was not a thin film completely independent of the NW-PP fabric, but a composite formed on the surface of the NW-PP fabric.

Figure 4 shows the C 1s, N 1s, and O 1s XPS spectra of NW-PP and CN-formed NW-PP fabrics. The C 1s peak in the NW-PP fabrics was observed at 284.5 eV, corresponding to the C–C/C–H bond. For the CN-formed NW-PP fabrics, deconvoluted peaks were mainly observed in the C 1s at 284.5 eV, 285.5 eV, and 288.5 eV, which were attributed to C–C/C–H, C–N, and C=O bonds, respectively. The N 1s peak could be deconvoluted to two peaks positioned at 398.7 eV and 399.6 eV, which were attributed to *sp*^2^ nitrogen atoms with two neighbors (=N-) and *sp*^2^ nitrogen atoms with three neighbors (–N<), respectively. The presence of the C=O bond was due to chemical adsorption in atmospheric conditions and confirmed by the O 1s peak. Usually, the oxygen peaks range between 529 and 535 eV. The peaks related to oxygen adsorption were observed between 530 and 535 eV, while the peaks below 530 eV represented lattice oxygen [29,30]. From these results, the decrease in C–C/C–H bonds in the C 1s indicated the removal of a methyl group (CH_3_ group) from the surface of the NW-PP fabrics, which was the result of easy interactions between the broken sites of the PP fabric and active nitrogen elements in the plasma. Considering the short plasma exposure time and SEM results, the C and N atoms seemed to bind to the sites broken by the high-energy plasma. This implied the formation of very thin C-N composites on the surface of the NW-PP fabrics.

Figure 5 shows contact angle images of DI water (polar liquid) and CH_2_I_2_ (non-polar liquid) droplets on the NW-PP and CN-formed NW-PP fabrics. The contact angles of the DI droplet on the NW-PP and CN-formed NW-PP fabrics were approximately 130° and 115°, respectively. Both samples showed a strong hydrophobicity against the polar liquid. However, the contact angle in the CN-formed NW-PP fabrics was slightly reduced compared to the NW-PP fabrics. This can be explained as the appearance of C=O bonds on the CN-formed NW-PP fabrics, which had a polar property, as shown in XPS results. In addition, the droplet of CH_2_I_2_ on NW-PP could only maintain its shape for a few seconds and then spread widely, while that on CN-formed NW-PP fabric was immediately spread. Thus, the CN-formed NW-PP fabric represented the full-wetting characteristic for a non-polar liquid, and this implied that as-formed CN composites on the fabrics played an important role in enhancing the wetting property for non-polar liquids. The commonly acceptable definition of hydrophobicity and hydrophilicity for water is based on 90°. It can be defined as a hydrophilic surface when the static contact angle (*θ*) for water is <90°, a hydrophobic surface when *θ* is >90°, and a superhydrophobic surface when *θ* is >150° [31]. Thus, the change in the contact angle was simply considered as the result of the competition between cohesion and wetting on the surface. In terms of the interaction between the water and the surface, it can be represented that the hydrophilic surface exhibited a strong affinity, while the hydrophobic surface showed little affinity for water solution. The intermolecular attraction was the source of the surface energy and was the sum of the contributions of polar and non-polar (or dispersive) components. The polar interaction between the surface and the solution was a Coulomb interaction between permanent dipoles or between permanent and induced dipoles, while the dispersive interaction was caused by temporal fluctuations in charge distribution in the surface and the solution. In addition, it is also important to identify the types of major biomolecules to determine the reactions between polymers and bacteria. Biological molecules are mainly composed of carbon and hydrogen atoms and can contain many different types of functional groups. The important functional groups in biological molecules are (-OH), methyl (-CH_3_), carbonyl (C=O), carboxyl (-COOH), amino (-NH_2_), sulfhydryl (-SH), and phosphate (-PO_4_) groups [32]. These functional groups play a crucial role in forming biomolecules such as proteins, carbohydrates, lips, and DNA. The functional groups can be divided as having polar and non-polar characteristics, which depends on their molecular structure and compositions. The methyl group is the only non-polar functional group, while other groups are polar groups. This meant that the methyl group represented a hydrophobic property, and the remaining group exhibited hydrophilic properties. The polar ($\gamma^{p}$) and dispersive surface energy ($\gamma^{d}$) on the surface of NW-PP fabrics were determined using the contact angles for DI ($\gamma_{L}^{p}$: 51 mJ/m^2^, $\gamma_{L}^{d}$: 21.9 mJ/m^2^, $\gamma_{L}$: 72.8 mJ/m^2^) and CH_2_I_2_ ($\gamma_{L}^{p}$: 0 mJ/m^2^, $\gamma_{L}^{d}$: 50.8 mJ/m^2^, $\gamma_{L}$: 50.8 mJ/m^2^) [20,33]. The calculations were performed by the Owen–Wendt method [20]. In this case, the polar elements of ordinary and CN-formed NW-PP fabrics were 2.01 mJ/m^2^ and 3.32 mJ/m^2^, respectively, showing little difference. Otherwise, the dispersive surface energy of the NW-PP fabrics after plasma treatment increased from 23.2 mJ/m^2^ to 59.2 mJ/m^2^, which in turn increased the total surface energy from 25.2 mJ/m^2^ to 62.5 mJ/m^2^.

Figure 6 shows the photographic images of the evolution of two different bacteria for the antibacterial test, and *S. aureus* and *K. pneumoniae* were used to verify the antibacterial effect of the CN-formed NW-PP fabrics. The images represented that the CN-formed NW-PP fabrics suppressed the growth of two different types of Gram-positive and Gram-negative bacteria. A brief description of the two types of bacteria is as follows. Gram-positive and Gram-negative bacteria are two different types of bacteria. Gram-positive bacteria appear blue or purple after Gram-staining in a laboratory test. They have thick cell walls (peptidoglycan cell walls). Gram-negative bacteria appear pink or red on staining and have thin walls (thinner peptidoglycan cell walls). They release different toxins and affect the body in different ways. *S. aureus* is a Gram-positive bacterium with a spherical shape and is a facultative anaerobe that can grow without the need for oxygen. *S. aureus* can become the main reason for opportunistic pathogens, skin infections, respiratory infections, and food poisoning. *K. pneumoniae* is a Gram-negative bacterium with a rod shape. This is a non-motile, encapsulated, lactose-fermenting, and facultative anaerobe bacteria. *K. pneumoniae* can cause destructive changes to human and animal lungs [34]. Table 1 summarizes the concentration changes of *S. aureus* and *K. pneumoniae* on the CN-formed NW-PP fabrics initially and after 18 h. The reduction rate (*R*) was calculated as follows:(1)$$R\;(\%)=\frac{B-A}{B}\times 100,$$
where *A* is the colony-forming unit (CFU) per mL of the control group, and *B* is the experimental group [21,35,36]. The reduction rate of the CN-formed NW-PP fabric was 89.0% and 91.6% for *S. aureus* (ATCC 6538) and *K. pneumoniae* (ATCC 4352), respectively, while that of the NW-PP fabric was 84.3% and 85.5%, respectively. In addition, the performance stability of the CN-formed NW-PP fabric was tested using samples left in the atmosphere for about 3 months. The reduction rate for the CN-formed NW-PP exhibited a similar value compared to previous samples. The reduction rate for the aged sample was 89.4% for *S. aureus* and 91.2% for *K. pneumoniae*.

The CN-formed NW-PP fabric showed an antibacterial reduction, which could be explained by the strong hydrophobicity of the fabrics. The CN-formed NW-PP fabric also demonstrated extreme wetting characteristics for non-polar liquids, as mentioned above, which could also have a significant impact on the respective antibacterial effects. Typically, the surface charge plays an important role in the interactions between materials. Bacteria can be adsorbed onto a material surface under electrostatic action, including Coulomb and van der Waals attractive forces. Usually, bacteria cells are negatively charged and coated with a peptidoglycan layer of sugars and amino acids [14]. Super-hydrophilic and -hydrophobic surfaces could yield low bacterial adhesion, affecting the antibacterial effects [9,14,15]. Recently, Yang et al. reported the antibacterial effects of a nanodiamond coating with C=O functional groups, and the appearance of C=O can also affect the antibacterial activity. They explained the antibacterial effect that the C=O bonds damaged the cell wall through the blockage and the repulsive electric force [37]. Thus, it was thought that the antibacterial activity for CN-formed NW-PP fabrics originated from the strong hydrophobicity due to the CH_3_ bond of NW-PP fabric, the enhanced wetting property due to CN bonds, and the antibacterial activity due to C=O bonds. Consequently, the proposed method is a one-step process that is simple, intuitive, and inexpensive. This is an eco-friendly method not requiring harmful precursors or toxic gases. Moreover, this method can be applied to most weak substrates because there is little damage to the sample. This suits mass production and provides an eco-friendly approach to producing antibacterial fabrics enhancing the antibacterial effect [9,14,15].

## 4. Conclusions

The non-woven polypropylene (NW-PP) fabric was plasma-treated to modify the surface properties using a modified DC-pulsed sputtering method with a roll-to-roll system. Compared to the commercially available sputtering gun, the configuration of the magnet was modified into alternately repeated N-polarity lines and S-polarity lines, which formed a magnetic field parallel to the substrate. After Ar/N_2_–C plasma treatments, there were no structural damages in the CN-formed NW-PP fabrics. It was confirmed that the combination of the modified sputtering gun and the DC-pulsed power could minimize damage to the surface. From XPS results, the chemical bonds on the surface were converted from C–C/C–H into C–C/C–H, C–N, and C=O bonds. CN-formed NW-PP exhibited a strong hydrophobicity for H_2_O (polar liquid) and full-wetting characteristics for CH_2_I_2_ (non-polar liquid). In addition, the CN-formed NW-PP fabric showed enhanced antibacterial properties compared to the NW-PP fabric. The reduction rate of the CN-formed NW-PP fabric was 89.0% and 91.6% for *Staphylococcus aureus* (ATCC 6538, Gram-positive) and *Klebsiella pneumoniae* (ATCC4352, Gram-negative), respectively, confirming that the CN layer represented antibacterial characteristics against both Gram-positive and Gram-negative bacteria. To sum up, it was concluded that the antibacterial activity for CN-formed NW-PP fabrics originated from the strong hydrophobicity due to CH_3_ bonds of NW-PP fabric, the enhanced wetting property due to CN bonds, and the antibacterial activity due to C=O bonds. However, it is thought that more research is needed to clarify the antibacterial effect. The proposed method is a one-step process that is simple, intuitive, and inexpensive, and does not involve any harmful precursors or toxic gases. The roll-to-roll process is suitable for the mass processing of flexible substrates, including polymers, fabrics, and polymer-like materials. Moreover, this eco-friendly method can be applied to the majority of weak substrates available because it does not cause any significant damage to the sample. Consequently, we believe that our study presents an effective eco-friendly method that allows the mass production of antibacterial fabrics.

## Figures and Tables

**Figure 1 polymers-15-02641-f001:**
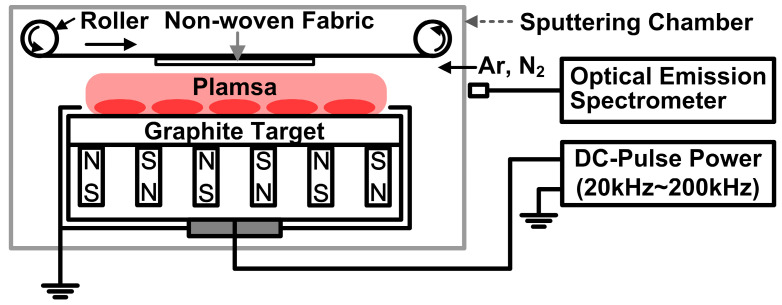
A schematic diagram of a DC-pulsed sputtering system for the formation of a carbon nitride (CN) layer on non-woven PP (NW-PP) fabrics.

**Figure 2 polymers-15-02641-f002:**
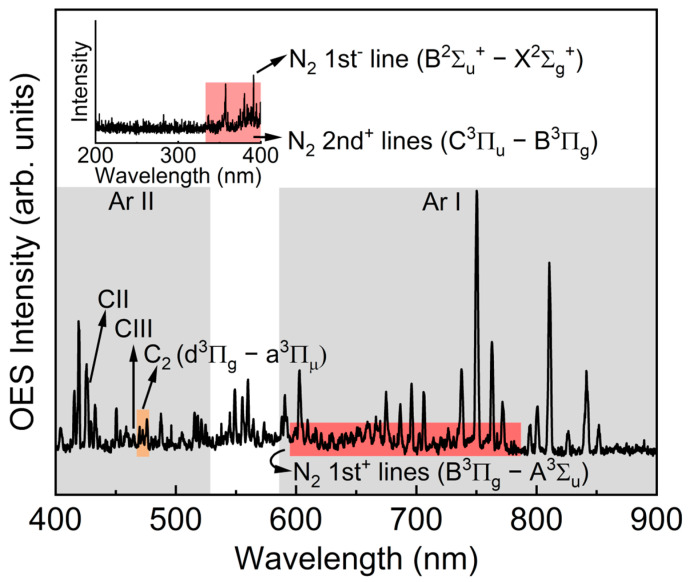
Optical emission spectrum of Ar/N_2_-C plasma generated by the DC-pulsed sputtering (Pressure: 0.23 Pa (Ar: 180 sccm, N_2_: 20 sccm), power: 450 W (voltage: 450 V, current: 1 A), square pulse frequency: 60 kHz (duty ratio of 50%)).

**Figure 3 polymers-15-02641-f003:**
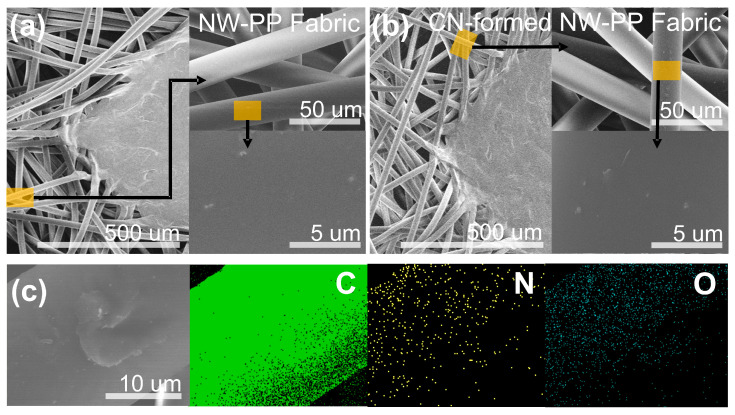
Morphological SEM images of the non-woven polypropylene (NW-PP) fabrics: (**a**) ordinary NW-PP fabric, (**b**) CN-formed NW-PP fabric, and (**c**) EDX elemental mapping images of C, N, and O for CN-formed NW-PP fabric.

**Figure 4 polymers-15-02641-f004:**
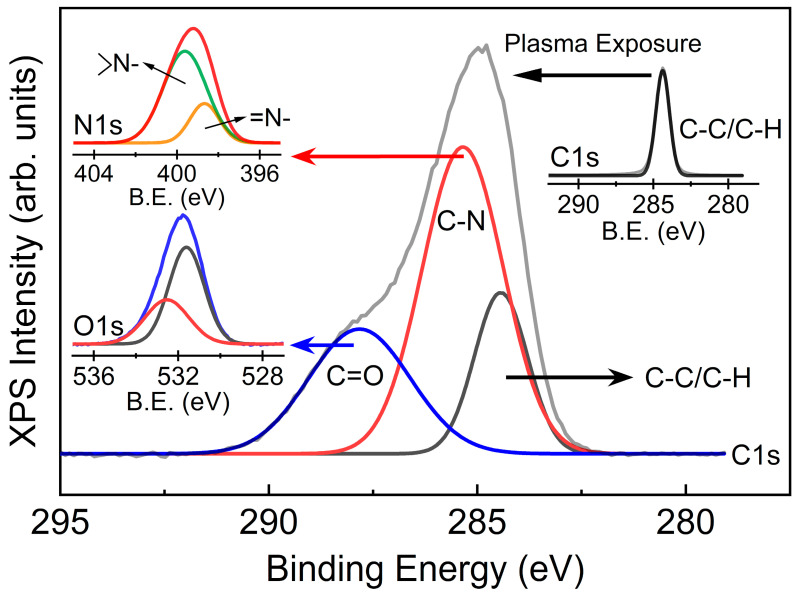
XPS C 1s spectra of non-woven polypropylene (NW-PP) and CN-formed NW-PP fabrics.

**Figure 5 polymers-15-02641-f005:**
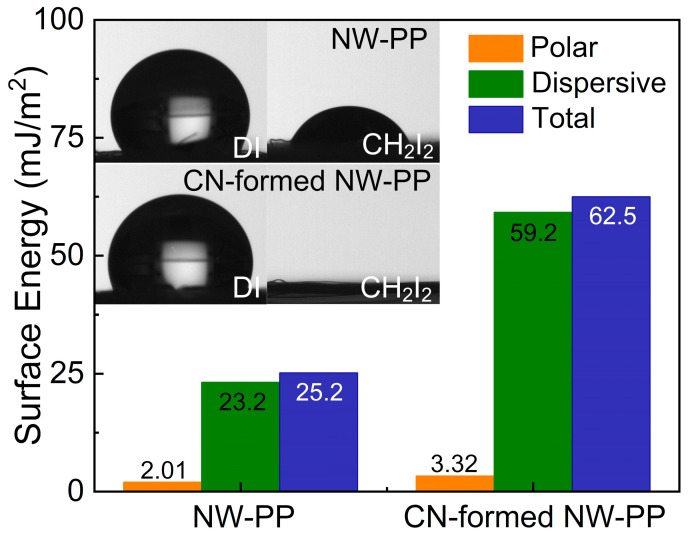
Analysis of the surface energy for the non-woven polypropylene (NW-PP) and CN-formed NW-PP fabric; (inset) photo-images of distilled H_2_O and CH_2_I_2_ droplets on the NW-PP and CN-formed NW-PP fabrics.

**Figure 6 polymers-15-02641-f006:**
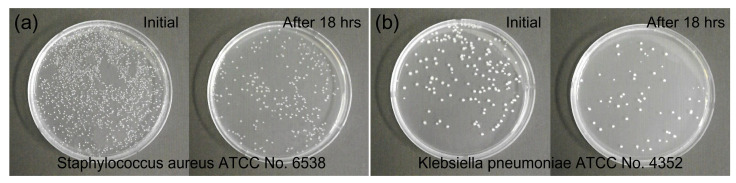
Photographs of bacteria evolutions for antibacterial test (KS K 0693 test): (**a**) *Staphylococcus aureus* (ATCC 6538) and (**b**) *Klebsiella pneumoniae* (ATCC 4352) on the CN-formed NW-PP fabrics. In (**a**,**b**), the left side shows the control groups and the right side shows the experimental groups after 18 h, which were incubated at a temperature of 37 °C for 18 h.

**Table 1 polymers-15-02641-t001:** The reduction rate (*R*) of *Staphylococcus aureus* (ATCC 6538) and *Klebsiella pneumoniae* (ATCC 4352) on the non-woven polypropylene (NW-PP) fabric and CN-formed NW-PP fabrics compared to the reference fabrics (Antibacterial test method: KS K 0693, 2016).

Bacteria Name	Bacteria Type	Testing Sample	Concentration (CFU/mL)		*R* (%)
			Initial	After 18 h.	
ATCC 6538 (*S. aureus*)	Gram-Negative	Reference	6.0 × 10^4^	2.8 × 10^6^	84.3
		NW-PP fabric		4.4 × 10^5^	
		Reference	8.9 × 10^4^	4.2 × 10^6^	89.0
		CN-formed NW-PP		4.6 × 10^5^	
ATCC 4352 (*K. pneumoniae*)	Gram-Positive	Reference	4.2 × 10^4^	2.0 × 10^6^	85.5
		NW-PP fabric		2.9 × 10^5^	
		Reference	3.2 × 10^4^	1.9 × 10^6^	91.6
		CN-formed NW-PP		1.6 × 10^5^	

## Data Availability

Not applicable.

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
