# Peer review of "Antibacterial Effects of a Carbon Nitride (CN) Layer Formed on Non-Woven Polypropylene Fabrics Using the Modified DC-Pulsed Sputtering Method"

_polymers, 2023, doi:10.3390/polym15122641_

Round 1
Reviewer 1 Report
This paper studies the influence of the carbon nitride (CN) coating on antibacterial properties of the PP fabric. The biocidal efficacy of modified NW-PP fabric has been tested against two strains of bacteria: Klebsiella pneumoniae (Gram-negative) and Staphylococcus aureus (Gram-positive).
The weak part of this paper is section Results and discussions. The results are described and analysed too quickly (in particularly the antibacterial properties). Furthermore, all discussions on the properties of the CN-modified PP fabric were reported separately where they should have been inter-related and discussed.
I am also surprised that in the discussion, the authors state that NW-PP and CN-formed NW-PP fabrics have a super-hydrophobic surfaces while the contact angles of the DI droplet on the NW-PP and CN-formed NW-PP fabrics were approximately 130° and 115°, respectively.
Super-hydrophobic surface is defined by having the static water contact angle above 150 ° and contact angle hysteresis less than 5 °.
With the above, I feel that the paper cannot be published in the present form.
I have other comments;
1) Experimental (Materials and Methods): Are the used DC-pulsed sputtering parameters previously optimized? Give the reference.
2) The characterization techniques and experimental conditions should be more detailed.
3) I am surprised that in the discussion, the first figure is talked about optical emission spectrum of Ar-N2 plasma which does not add useful information in this study.
4) It will be interesting to explore the effects of the CN film thickness on the functionalization and antibacterial properties. Given that the authors can easily modify the parameters on the reactor this should be a straight forward experiment that will add much more depth to the study.
5) In introduction the authors make known that “the activated carbon (AC) has a larger surface area due to its porous structure, a property which can efficiently promote antibacterial effects”. Is the NC coatings have a porous structure? Is the coating a continuous layer? The authors should clarify the discussion in this part to more accurately reflect the data. In my opinion, some AFM images can be provided.
6) The contact angle hysteresis of these surfaces must be evaluated.
7) In section antibacterial activity, the authors present only the reduction of bacteria (R). It is necessary that the authors show some images of their antibacterial test to warrant the results presented in table 1. In my opinion, it is also required to perform other antibacterial tests (agar-plate diffusion test, Inhibition tests …)
8) Tables 1: the authors should add the Standard deviation. This information is essential to conclude on the significance of the reduction rate (R).
9) The main comment relating to the experimental results of antibacterial effect is the lack of detail in their discussion. The authors explain the antibacterial effect by only the wetting properties of CN-formed NW-PP fabric. The authors should clarify the discussion in this part to more accurately reflect the experimental data.
Does not the treatment change the mechanical properties of fabric?
10) The stability and durability of antibacterial effectiveness should be clarified by experiment since the authors hope to obtain fabrics with durable antibacterial properties.
11) It would be beneficial to include comparison with synthetic fabric or blend of synthetic and natural fibers to see if the described method can be used in wide range of materials.
12) The paper requires a few language corrections.
Reviewer 2 Report
The paper presents new results for antibacterial effects of CN layers of non-woven PP fabrics. There are some shortcomings which must be removed prior to publication.
Introduction: Cytocompatibility of amorphous hydrogenated carbon nitride films deposited by an atmospheric dielectric barrier discharge plasma was investigated by Majumdar et al., J. Appl. Phys. 104, 074702 (2008); DOI: 10.1063/1.2990054 . Atmospheric pressure plasmas have several advantageous compared to low pressure plasmas. A brief discussion related to this work is required.
Line 32: why “outdoors”? Delete.
Line 58: add duty cycle and/or pulse length of the discharge.
Line 61: replace by “0.23 Pa”.
Line 69: “The surface morphology of the CN-formed”? Is not a sentence. Delete?
Line 76: “requesting” or appointing?
Line 88: replace “unionized” by “neutral”
Figure 2: how was optical emission spectroscopy (OES) performed? What kind of spectrograph, grating, resolution, detector, etc. It is a pretty poor spectrum. Assignments are wrong, use NIST data tables, e.g., https://www.nist.gov/pml/atomic-spectra-database , to identify optical emission line. For example, prominent Ar I lines also occur in the range of 400-450 nm. What do you want to learn from it?
Figure 3, caption: (b) “CN-coated” ?
Line 109: kinetic energy of 600 eV? How come, you only apply 450 V, so, the bombarding energy is 450 eV or less.
Line 114: typical “information” depth ?
Lies 124/125: “=N” is usually used for double bonds with a single atom. In my opinion “-N-“ appears better to indicate 2 neighbours. “-N<” looks pretty strange, probably okay …
Line 133: replace “particles” by “atoms”.
Line 201: “inexpensive” is rather relative. Atmospheric pressure plasmas, see Majumdar et al., are much less expensive.
Should be improved
Reviewer 3 Report
The manuscript addresses the relevant topic of plasma-based surface modification of non-woven polymeric materials.
Unfortunately, I can not recommend this manuscript for publication for the following reason:
A key experiment of this study is the antibacterial test. Regarding this matter the authors report, that the results were requested from an external agency. Numerical data are presented but no experimental details on the test procedure are revealed in the text. Only a citation is given. Moreover, the results are compared to a ‘reference fabric’ which is not specified.
Nevertheless, I would like to add some comments for a possible resubmission:
The emission spectrum of the plasma is nicely interpreted but the results should be linked to the main story of polymer surface modification.
The interpretation of contact angle data should consider the fact that this is a ‘composite surface’.
Usually, the range of a plasma treatment is very limited in a porous material like this. The problem should be clearly addressed in the discussion.
Round 2
Reviewer 1 Report
Although most of the points raised by the reviewer have been addressed satisfactorily, the paper still needs some minor corrections and clarifications. I'd ask the authors to take into account the points raised by Reviewer 3 in their revision.
Reviewer 2 Report
The paper is improved and now ready for publication.
Reviewer 3 Report
The previous concerns of the reviewer were largely considered in the revised version.